# Single microwave-photon detector using an artificial Λ-type three-level system

Kunihiro Inomata[1,*], Zhirong Lin[1,*], Kazuki Koshino[2], William D. Oliver[3,4], Jaw-Shen Tsai[1,5], Tsuyoshi Yamamoto[6] & Yasunobu Nakamura[1,7]

Single-photon detection is a requisite technique in quantum-optics experiments in both the optical and the microwave domains. However, the energy of microwave quanta are four to five orders of magnitude less than their optical counterpart, making the efficient detection of single microwave photons extremely challenging. Here we demonstrate the detection of a single microwave photon propagating through a waveguide. The detector is implemented with an impedance-matched artificial Λ system comprising the dressed states of a driven superconducting qubit coupled to a microwave resonator. Each signal photon deterministically induces a Raman transition in the Λ system and excites the qubit. The subsequent dispersive readout of the qubit produces a discrete 'click'. We attain a high single-photon-detection efficiency of $0.66 \pm 0.06$ with a low dark-count probability of $0.014 \pm 0.001$ and a reset time of $\sim 400$ ns. This detector can be exploited for various applications in quantum sensing, quantum communication and quantum information processing.

[1] RIKEN Center for Emergent Matter Science (CEMS), Wako 351-0198, Saitama, Japan. [2] College of Liberal Arts and Sciences, Tokyo Medical and Dental University, Ichikawa 272-0827, Chiba, Japan. [3] MIT Lincoln Laboratory, Lexington, Massachusetts 02420, USA. [4] Departent of Physics, Massachusetts Institute of Technology, Cambridge, Massachusetts 02139, USA. [5] Department of Physics, Tokyo University of Science, Shinjuku-ku, Tokyo 162-8601, Japan. [6] NEC IoT Device Research Laboratories, Tsukuba 305-8501, Ibaraki, Japan. [7] Research Center for Advanced Science and Technology (RCAST), The University of Tokyo, Meguro-ku, Tokyo 153-8904, Japan. * These authors contributed equally to this work. Correspondence and requests for materials should be addressed to K.I. (email: kunihiro.inomata@riken.jp).

Single-photon detection is essential to many quantum-optics experiments, enabling photon counting and its statistical and correlational analyses[1]. It is also an indispensable tool in many protocols for quantum communication and quantum information processing[2–5]. In the optical domain, various kinds of single-photon detectors are commercially available and commonly used[1,6]. However, despite the latest developments in nearly-quantum-limited amplification[7,8] and homodyne measurement for extracting microwave photon statistics[9], the detection of a single microwave photon in an itinerant mode remains a challenging task due to its correspondingly small energy. Meanwhile, the demand for such detectors is rapidly increasing, driven by applications involving both microwave and hybrid optical-microwave quantum systems. In this article we demonstrate an efficient and practical single microwave-photon detector based on the deterministic switching in an artificial Λ-type three-level system implemented using the dressed states of a driven superconducting quantum circuit. The detector operates in a time-gated mode and features a high quantum efficiency $0.66 \pm 0.06$, a low dark-count probability $0.014 \pm 0.001$, a bandwidth $\sim 2\pi \times 16$ MHz, and a fast reset time $\sim 400$ ns. It can be readily integrated with other components for microwave quantum optics.

Our detection scheme carries several advantages compared with previous proposals. It uses coherent quantum dynamics, which minimizes energy dissipation on detection and allows for rapid resetting with a resonant drive, in contrast to schemes that involve switching from metastable states of a current-biased Josephson junction into the finite voltage state[10–12]. Moreover, our detection scheme does not require any temporal shaping of the input photons, nor precise time-dependent control of system parameters adapted to the temporal mode of the input photons, in contrast to the photon-capturing experiments[13–15]. Temporal mode mismatch of the photons also limits the maximum efficiency in the recently demonstrated single-photon detection using a transmon qubit in a three-dimensional (3D) cavity[16]. Finally, our scheme also achieves a high efficiency without cascading many devices[10,17].

The operating principle of the detector fully employs the elegance of waveguide quantum electrodynamics, which has recently attracted significant attention in various contexts surrounding photonic quantum information processing[18–21]. When electromagnetic waves are confined and propagate in a one-dimensional (1D) mode, their interaction with a quantum emitter/scatterer is substantially simplified and enhanced compared with 3D cases. These advantages result from the natural spatial-mode matching of the emitter/scatterer with a 1D mode and its resulting enhancement of quantum interference effects. Remarkable examples are the perfect extinction of microwave transmission for an artificial atom coupled to a 1D transmission line[22,23], the photon-mediated interaction between two remote atoms coupled to a 1D transmission line[24], and the perfect absorption— and thus 'impedance matching'— of a Λ-type three-level system terminating a 1D transmission line[25,26]. In the latter system, the incident photon deterministically induces a Raman transition, which switches the state of the Λ system[25,27]. This effect has recently been demonstrated in both the microwave and optical domains[26,28], indicating its potential for photon

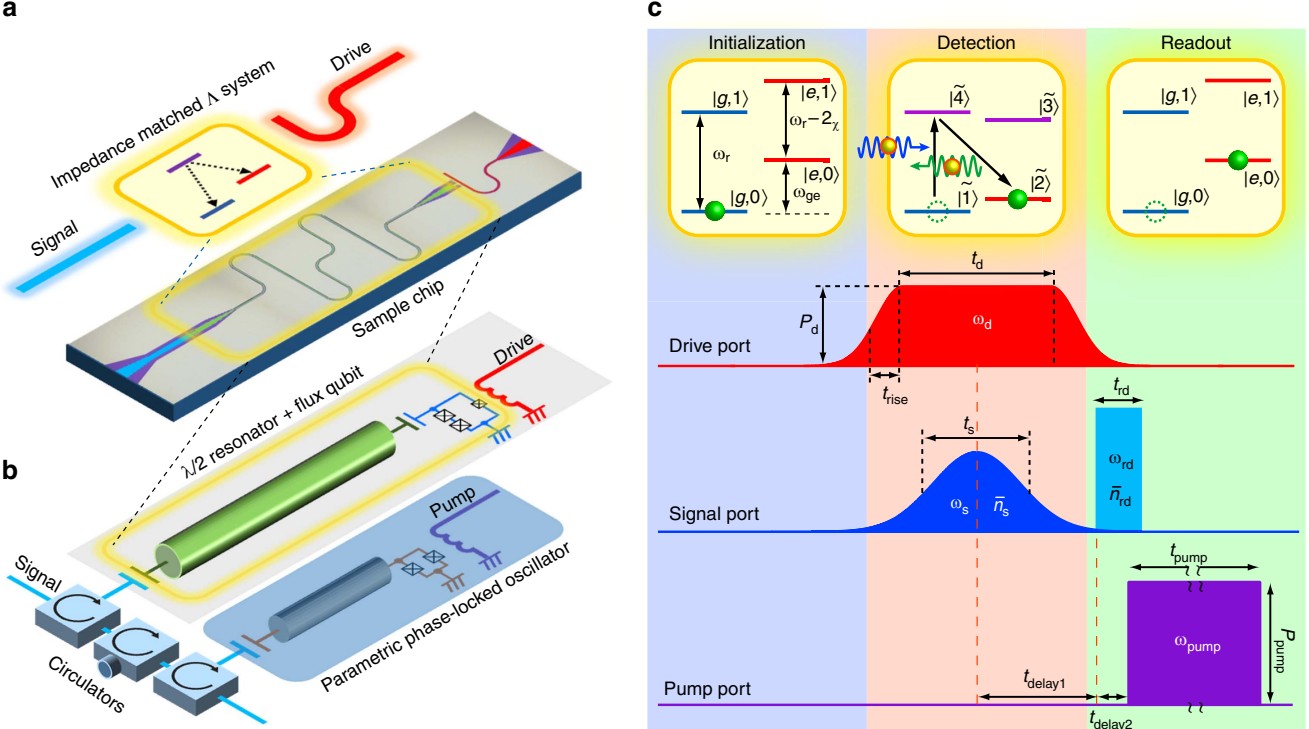

**Figure 1 | Experimental set-up and pulse sequence. (a)** Image of the sample chip containing a flux qubit and a superconducting microwave resonator coupled capacitively and operated in the dispersive regime. For certain proper conditions of the qubit drive, the coupled system functions as an impedance-matched Λ-type three-level system. **(b)** Schematic of the itinerant microwave-photon detector consisting of the coupled system and connected to a parametric phase-locked oscillator (PPLO) via three circulators in series. The circuit has three input ports: signal, qubit drive, and pump for the PPLO. **(c)** Energy-level diagram of the coupled system and the pulse sequence for single-photon detection. The system is first prepared in the ground state. During the detection stage, we concurrently apply the drive and signal pulses. The drive is parameterized to fulfil the impedance-matched condition such that a signal photon (blue arrow) induces a deterministic Raman transition. A downconverted photon (green arrow) is emitted in the process and discarded. In the readout stage, we detect the qubit excited state nondestructively by sending a qubit readout pulse. The qubit-state-dependent phase shift in the reflected pulse is discriminated by the PPLO. Detailed parameters of the pulse sequence are provided in Methods.

detection[29] as well as for implementing deterministic entangling gates with photonic qubits[30].

## Results

**Implementation of a single microwave-photon detector.** Our device consists of a superconducting flux qubit capacitively and dispersively coupled to a microwave resonator (Fig. 1b and ref. 31; also see Supplementary Note 1 and Supplementary Fig. 1 for the details of the device). With a proper choice of the qubit drive frequency $\omega_d$ and power $P_d$, the system functions as an impedance-matched $\Lambda$ system with identical radiative decay rates from its upper state to its two lower states (Fig. 1a)[25,26]. The qubit–resonator coupled system is connected to a parametric

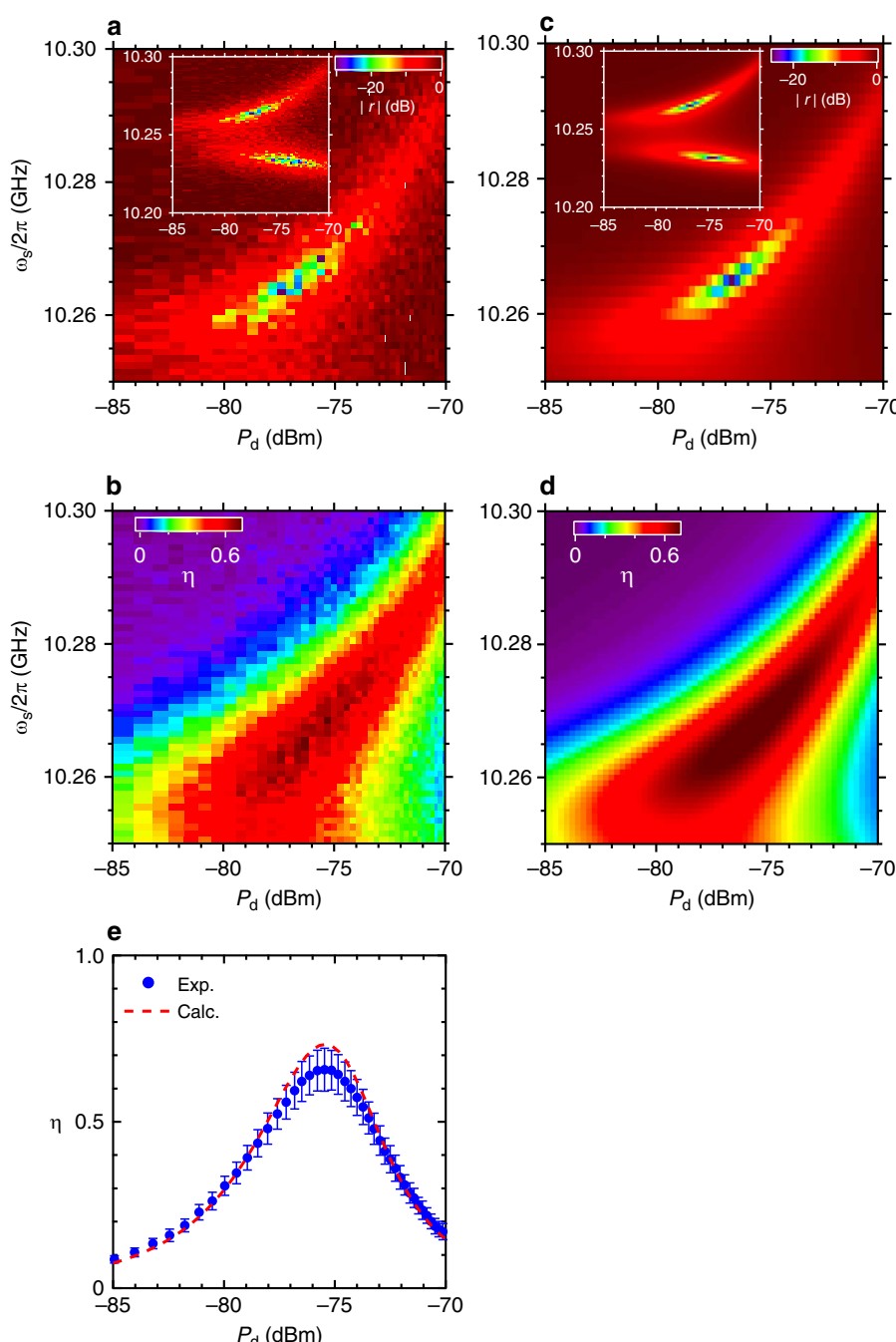

**Figure 2 | Impedance matching and itinerant microwave-photon detection.** (**a**) Amplitude of the reflection coefficient $|r|$ of the input signal pulse with mean photon number $\bar{n}_s \sim 0.1$ as a function of the qubit drive power $P_d$ and the signal frequency $\omega_s$. The PPLO is not activated during this measurement. The impedance-matched region is resolved (dark-blue point), where the input microwave photon is absorbed almost completely. In the inset, we also observe another dip in $|r|$, corresponding to the Raman transition of $|\tilde{1}\rangle \rightarrow |\tilde{3}\rangle \rightarrow |\tilde{2}\rangle$. Microwave power levels stated in this article are referred to the value at the corresponding ports on the sample chip. (**b**) Detection efficiency $\eta$ of an itinerant microwave photon. The efficiency reaches its maximum at the impedance-matched point, where the Raman transition of $|\tilde{1}\rangle \rightarrow |\tilde{4}\rangle \rightarrow |\tilde{2}\rangle$ takes place. (**c,d**) Theoretical predictions corresponding to **a** and **b**. (**e**) Cross-sections of (**b**) (blue dots) and (**d**) (red dashed line) at $\omega_s/2\pi = 10.268$ GHz. The error bars are due to the uncertainty in the input power calibration (see Supplementary Note 5 and Supplementary Fig. 5 for the details of the input power calibration).

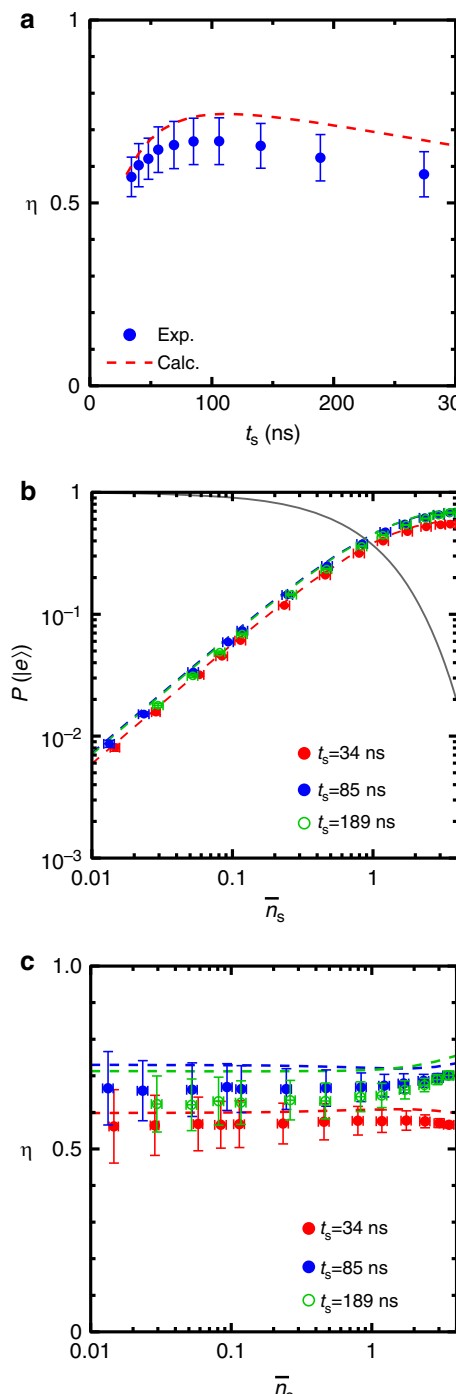

**Figure 3 | Optimization of the efficiency.** (**a**) Single-photon-detection efficiency $\eta$ as a function of the signal pulse length $t_s$. The mean photon number $\bar{n}_s$ for the weak-coherent signal pulse is $\sim 0.1$. (**b**) Probability of the qubit excitation by single microwave photons, $P(|e\rangle)$, as a function of $\bar{n}_s$. Dashed lines represent theoretical predictions and the solid line indicates probability for the signal pulse being in the vacuum state, $P(0) = \exp(-\bar{n}_s)$. The statistical error in $P(|e\rangle)$ is smaller than the dot size. (**c**) $\eta$ calculated from $P(|e\rangle)$ and $P(0)$ in **b** as a function of $\bar{n}_s$. Dashed lines indicate theoretical predictions. In all the plots, the error bars both in $\bar{n}_s$ and $\eta$ are due to the uncertainty in the input power calibration.

phase-locked oscillator (PPLO), which enables fast and non-destructive qubit readout (ref. 32; also see Supplementary Notes 1 and 2, and Supplementary Fig. 2 for the details of the device and the experimental set-up).

Figure 1c shows the level structure of the qubit–resonator system and the protocol for the single-photon detection. We label the energy levels $|q, n\rangle$ and their eigenfrequencies $\omega_{|q,n\rangle}$, where $q = \{g, e\}$ and $n = \{0, 1, \cdots\}$, respectively, denote the qubit state and the photon number in the resonator. In the dispersive coupling regime, the qubit–resonator interaction renormalizes the eigenfrequencies to yield $\omega_{|g,n\rangle} = n\omega_r$ and $\omega_{|e,n\rangle} = \omega_{ge} + n(\omega_r - 2\chi)$, where $\omega_{ge}$ and $\omega_r$ are the renormalized frequencies of the qubit and the resonator, respectively, and $\chi$ is the dispersive frequency shift of the resonator due to its interaction with the qubit. Only the lowest four levels with $n = 0$ or 1 are relevant here.

We prepare the system in its ground state $|g, 0\rangle$ (Fig. 1c, Initialization) and apply a drive pulse to the qubit (Fig. 1c, Detection). In a frame rotating at $\omega_d$, the level structure becomes nested, that is, $\omega_{|g,0\rangle} < \omega_{|e,0\rangle} < \omega_{|e,1\rangle} < \omega_{|g,1\rangle}$, for $\omega_d$ in the range $\omega_{ge} - 2\chi < \omega_d < \omega_{ge}$ (refs 25, 26). On the plateau of the drive pulse, the lower-two levels $|g, 0\rangle$ and $|e, 0\rangle$ (higher-two levels $|g, 1\rangle$ and $|e, 1\rangle$) hybridize to form dressed states $|\tilde{1}\rangle$ and $|\tilde{2}\rangle$ ($|\tilde{3}\rangle$ and $|\tilde{4}\rangle$). Under a proper choice of $P_d$, the two radiative decay rates from $|\tilde{4}\rangle$ (or $|\tilde{3}\rangle$) to the lowest-two levels become identical. Thus, an impedance-matched $\Lambda$ system comprising $|\tilde{1}\rangle$, $|\tilde{2}\rangle$ and $|\tilde{4}\rangle$ (alternatively, $|\tilde{1}\rangle$, $|\tilde{2}\rangle$ and $|\tilde{3}\rangle$) is realized. An incident single microwave photon (Gaussian envelope, length $t_s$), synchronously applied with the drive pulse through the signal port and in resonance with the $|\tilde{1}\rangle \rightarrow |\tilde{4}\rangle$ transition, deterministically induces a Raman transition, $|\tilde{1}\rangle \rightarrow |\tilde{4}\rangle \rightarrow |\tilde{2}\rangle$, and is downconverted to a photon at the $|\tilde{4}\rangle \rightarrow |\tilde{2}\rangle$ transition frequency. This process is necessarily accompanied by an excitation of the qubit.

Finally, we adiabatically switch off the qubit drive and dispersively read out the qubit state (Fig. 1c, Readout). We apply a readout pulse with the frequency $\omega_{rd} = \omega_r - 2\chi = \omega_{|e,1\rangle} - \omega_{|e,0\rangle}$ through the signal port, which, on reflection at the resonator, acquires a qubit-state-dependent phase shift of 0 or $\pi$. This phase shift is detected by the PPLO with high fidelity: in the present set-up, the readout fidelity of the qubit is $\sim 0.9$, which is primarily limited by qubit relaxation before readout[32].

**Demonstration of single microwave-photon detection.** We first determine the operating point where the $\Lambda$ system deterministically absorbs a signal photon. We simultaneously apply a drive pulse of length $t_d = 178$ ns and a signal pulse of length $t_s = 85$ ns, and proceed to measure the reflection coefficient $|r|$ of the signal pulse as a function of the drive power $P_d$ and the signal frequency $\omega_s$ (Fig. 2a). The signal pulse is in a weak coherent state with mean photon number $\bar{n}_s \sim 0.1$. A pronounced dip with a depth of $< -25$ dB is observed in $|r|$ at $(P_d, \omega_s/2\pi) = (-76$ dBm, 10.268 GHz), in close agreement with theory (Fig. 2c). The dip indicates a near-perfect absorption condition, that is, impedance matching, where the reflection of the input microwave photon vanishes due to destructive self-interference. Correspondingly, a deterministic Raman transition of $|\tilde{1}\rangle \rightarrow |\tilde{4}\rangle \rightarrow |\tilde{2}\rangle$ is induced, and the qubit state is flipped.

To obtain a 'click' corresponding to single-photon detection, we read out the qubit state by using the PPLO immediately after the Raman transition. Before initiating readout, the drive pulse is turned off to suppress unwanted Raman transitions induced by the readout pulse, for example, $|\tilde{2}\rangle \rightarrow |\tilde{3}\rangle \rightarrow |\tilde{1}\rangle$. We repeatedly apply the pulse sequence in Fig. 1c $10^4$ times and evaluate the single-photon-detection efficiency $\eta \equiv P(|e\rangle)/[1 - P(0)]$, where $P(|e\rangle)$ and $P(0) \equiv \exp(-\bar{n}_s)$ are the probabilities for the qubit being in the excited state and the signal pulse being in the vacuum state, respectively. We emphasize that the detection efficiency here is defined with respect to the mean photon number in the

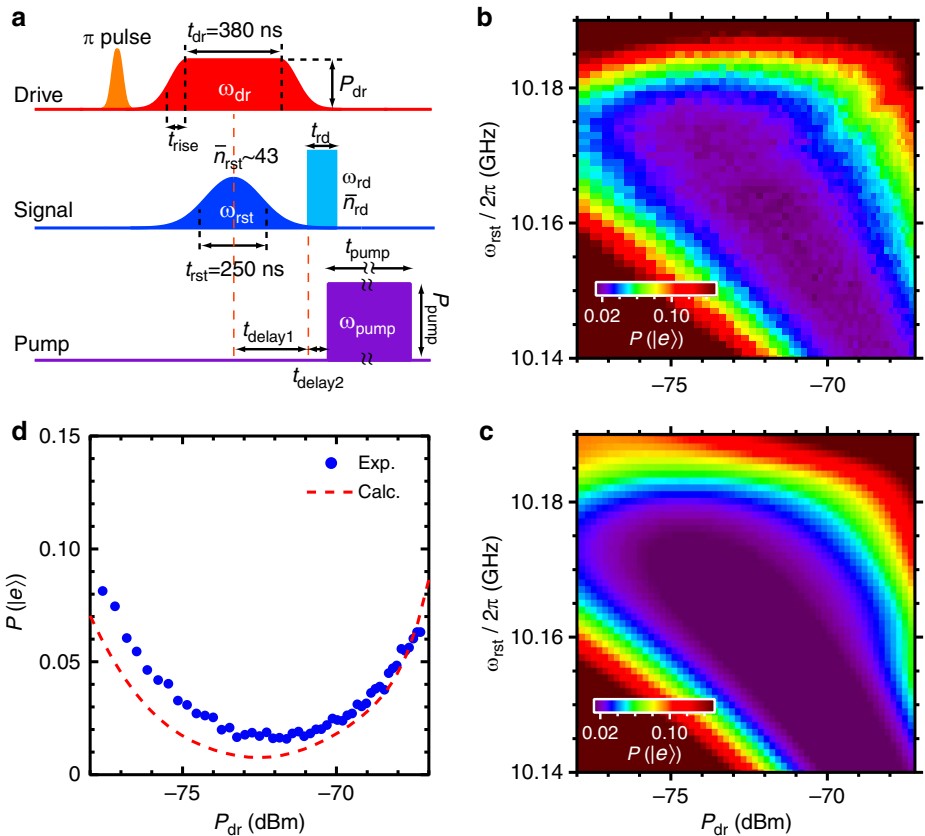

**Figure 4 | Demonstration of the fast reset protocol. (a)** Pulse sequence used to evaluate the reset efficiency. The initial $\pi$-pulse mimics a single-photon detection and excites the qubit. During the reset stage, a drive pulse and a reset pulse with the mean photon number of $\bar{n}_{rst} \sim 43$ are concurrently applied, inducing an inverse Raman transition: $|\tilde{2}\rangle \rightarrow |\tilde{3}\rangle \rightarrow |\tilde{1}\rangle$. The remaining population in the $|e\rangle$ state is then detected. **(b)** Population of the qubit excited state after the reset operation $P(|e\rangle)$, as a function of the reset-pulse frequency $\omega_{rst}$ and the drive-pulse power $P_{dr}$. **(c)** Theoretical prediction for **(b)** with no free parameters. **(d)** Cross sections of **(b)** (blue dots) and **(c)** (red dashed line) at $\omega_{rst}/2\pi = 10.162$ GHz.

propagating signal pulses. Figure 2b depicts $\eta$ as a function of $P_d$ and $\omega_s$. The dark-count probability of the detector—mainly caused by the nonadiabatic qubit excitation due to the drive pulse and the imperfect initialization—is subtracted when evaluating $\eta$ (see Supplementary Note 3 and Supplementary Fig. 3 for the details of the dark count in the detector). We observe that $\eta$ is maximized at the dip position in Fig. 2a in accordance with the impedance-matching condition. We also confirm that the result agrees with numerical calculations based on the parameters determined independently (Fig. 2d). The maximum value, $\eta = 0.66 \pm 0.06$, is obtained at $(P_d, \omega_s/2\pi) = (-75.5 \text{ dBm}, 10.268 \text{ GHz}; \text{ Fig. 2e})$. The efficiency exceeds 0.5 over a signal-frequency range of $\sim 20 \text{ MHz}$, which is comparable to the bandwidth of the detector, $\kappa/2\pi \sim 16 \text{ MHz}$ (see Supplementary Note 4 and Supplementary Fig. 4 for the details of the time constant of the impedance-matched $\Lambda$ system). $\bar{n}_s$ is maintained near 0.1 in the measurement, implying that $\sim 0.5\%$ of the weak-coherent signal pulses contain multiple photons. Our detector also responds to multi-photon pulses, as do many photodetectors, but it cannot discriminate them from single-photon pulses. The efficiency $\eta$ includes those counts. We theoretically confirm that our detector also works for other signal-pulse shapes such as rectangular and exponential decay[29].

**Optimization of detection efficiency.** In Fig. 3a, we plot efficiency $\eta$ as a function of the signal pulse length $t_s$. Here, we fix $\omega_s$ and $P_d$ at the values which maximize $\eta$ in Fig. 2e. The drive pulse duration $t_d$ is set to be $t_d = 1.5 t_s + 50 \text{ ns}$, which empirically maximizes $\eta$ at each $t_s$. We observe that $\eta$ is a non-monotonic function of $t_s$ and attains a maximum at $t_s = 85 \text{ ns}$. The initial increase of $\eta$ at short $t_s$ is due to the narrowing of the signal bandwidth resulting in an improved overlap with the detection bandwidth. The characteristic response time of the impedance-matched $\Lambda$ system is estimated to be $2/\kappa = 20 \text{ ns}$ in terms of the voltage amplitude. The shortest signal pulse length 34 ns in Fig. 3a is comparable to this. For longer $t_s$, the qubit relaxation limits $\eta$ (ref. 29). Next, we examine how the photon detector behaves when $\bar{n}_s$ in the signal pulse is varied. Figure 3b shows $P(|e\rangle)$ as a function of $\bar{n}_s$ for fixed signal pulse lengths at $t_s = 34, 85$, and 189 ns. $P(|e\rangle)$ increases linearly with $\bar{n}_s$ as expected. Moreover, the observed $P(|e\rangle)$ agree very well with the theoretically predicted values (dashed lines) based on the independently calibrated qubit lifetime and input signal power (Supplementary Note 5). Figure 3c shows the photon detection efficiency $\eta$ calculated from $P(|e\rangle)$ and $P(0)$ in Fig. 3b. The detection efficiencies stay constant for $\bar{n}_s \lesssim 1$ regardless of the pulse lengths. This validates the determination of $\eta$ in our measurements using signal pulses in weak coherent states. For $\bar{n}_s > 1$, $\eta$ slightly depends on $\bar{n}_s$ because of the possibility to drive multiple Raman transitions.

**Demonstration of a fast reset protocol.** After a single-photon-detection event, the qubit remains in the excited state until it spontaneously relaxes to the ground state, which leads to a relatively long dead time of the detector. However, our coherent

approach allows us to implement a fast reset protocol (Fig. 4a): in conjunction with the drive pulse that forms the $\Lambda$ system, we apply a relatively strong reset pulse through the signal port, which induces an inverse Raman transition, $|\tilde{2}\rangle \rightarrow |\tilde{3}\rangle \rightarrow |\tilde{1}\rangle$. We optimize the drive-pulse power $P_{\rm dr}$ and the reset-pulse frequency $\omega_{\rm rst}$ (see Methods section) such that the resulting qubit excitation probability $P(|e\rangle)$ is minimized (Fig. 4b). At the optimal reset point $(P_{\rm dr}, \omega_{\rm rst}/2\pi) = (-72.1\ \text{dBm}, 10.162\ \text{GHz})$, $P(|e\rangle)$ attains a minimum value $0.017 \pm 0.002$, equivalent to the value $0.016 \pm 0.001$ obtained in the absence of the initial $\pi$-pulse used to mimic a photon absorption event. Without a reset pulse, we obtain $P(|e\rangle) = 0.490 \pm 0.010$. A comparison of the two results indicates that the reset pulse is highly efficient. However, the reset pulse results in a twice-larger occupation of the qubit excited state compared with the value $0.008 \pm 0.001$ obtained through equilibration. This indicates a small probability of unwanted non-adiabatic excitation due to the drive pulse during the reset protocol. Finally, we demonstrate microwave photon detection combined with the fast reset protocol. We apply the drive and the signal pulses (the same conditions as in the measurement in Fig. 2b) after the reset protocol and readout the qubit. We achieve $\eta = 0.67 \pm 0.06$, consistent with the maximum value of $\eta$ in Fig. 2e. This indicates that the reset protocol does not affect subsequent detection efficiency. The time-gated operation with the reset protocol can be repeated at a rate exceeding 1 MHz (see Methods section).

## Discussion

For the moment, the detection efficiency of this detector is limited by the relatively short qubit relaxation time $T_1 \sim 0.7\ \mu\text{s}$. Nonetheless, our theoretical work indicates that efficiencies reaching $\sim 0.9$ are readily achievable with only a modest improvement of the qubit lifetime[29]. An extension from time-gated-mode to continuous-mode operation is also possible[33].

## Methods

**Protocol for the single-photon detection.** The drive frequency is set at $\omega_{\rm d} = \omega_{\rm ge} - \delta\omega$, where $\delta\omega = 2\pi \times 49\ \text{MHz}\ (<2\chi)$ is the detuning from the qubit energy and is fixed through all the experiments. The drive pulse is synchronized with the signal pulse, which has a Gaussian envelope with a length $t_{\rm s}$ corresponding to its full width at half maximum in its voltage amplitude (Fig. 1c). The duration $t_{\rm d}$ of the drive pulse is optimized as $t_{\rm d} = 1.5t_{\rm s} + 50\ \text{ns}$ so that the signal pulse is completely covered by the drive pulse and is efficiently absorbed by the $\Lambda$ system. To suppress unwanted nonadiabatic qubit excitations, the rising and falling edges of the drive-pulse envelope are smoothed by a Gaussian function with full width at half maximum of $2t_{\rm rise} = 30\ \text{ns}$ in its voltage amplitude.

The readout pulse (with frequency $\omega_{\rm rd} = \omega_{\rm r} - 2\chi = 2\pi \times 10.187\ \text{GHz}$, length $t_{\rm rd} = 60\ \text{ns}$, and mean photon number $\bar{n}_{\rm rd} \sim 10$) is applied after a delay of $t_{\rm delay1} = t_{\rm d}/2 + t_{\rm rise}$ from the centre of the drive and signal pulses. The reflected readout pulse works as a locking signal for the PPLO output phase, and the pump pulse (with frequency $\omega_{\rm pump} = 2\omega_{\rm rd}$, length $t_{\rm pump} = 400\ \text{ns}$, and power $P_{\rm pump} \sim -60\ \text{dBm}$) is applied after $t_{\rm delay2} = 40\ \text{ns}$. The parametric oscillation signal with either 0 or $\pi$ phase is output from the PPLO during the application of the pump pulse, and a data acquisition time of $\sim 100\ \text{ns}$ is required to extract the phase.

**Optimization of the reset protocol.** We first apply a $\pi$ pulse of length 6 ns to directly excite the qubit from the $|g, 0\rangle$ to the $|e, 0\rangle$ state (Fig. 4a). Then, we apply the drive and reset pulses to induce the $|\tilde{2}\rangle \rightarrow |\tilde{3}\rangle \rightarrow |\tilde{1}\rangle$ transition. To find the operating point which maximizes the reset efficiency, we swept the frequency $\omega_{\rm rst}$ of the reset pulse and the drive power $P_{\rm dr}$. After fixing $\omega_{\rm rst}$ and $P_{\rm dr}$, we adjust the drive pulse length $t_{\rm dr}$, and the mean photon number in the reset pulse $\bar{n}_{\rm rst}$ to minimize $P(|e\rangle)$. Finally we measure $P(|e\rangle)$ as a function of $\omega_{\rm rst}$ and $P_{\rm dr}$ using the reset protocol with optimized parameters. Parameters for the readout and pump pulses are the same as the ones in Fig. 1c.

It takes 410 ns to reset the system and 208 ns to detect a single photon for $t_{\rm s} = 85\ \text{ns}$. Both of the durations are determined by the drive pulse widths including $2t_{\rm rise} = 30\ \text{ns}$. The qubit readout is completed by accumulating data for 100 ns after $t_{\rm delay2} = 40\ \text{ns}$. The period of the single-photon detection including the reset protocol is $\sim 760\ \text{ns}$, which allows a photon counting rate of $\sim 1.3\ \text{MHz}$.

**Data availability.** The data that support the findings of this study are available from the corresponding author upon request.

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

## Acknowledgements

This work was partially supported by JSPS KAKENHI (Grant Number 25400417, 26220601, 15K17731), ImPACT Program of Council for Science and the NICT Commissioned Research.

## Author contributions

K.K., T.Y., Y.N., K.I. and Z.R.L. conceived the experiment. K.I. designed and fabricated the qubit device. T.Y. designed PPLO, which was fabricated at the group of W.D.O. Z.R.L characterized the PPLO. K.I. and Z.R.L performed the measurement and data analysis. K.K. developed the theory and performed the numerical simulations. K.I. prepared the manuscript. All authors contributed to the discussion of the results and helped in editing the manuscript.

## Additional information

**Competing financial interests:** The authors declare no competing financial interests.

