## [Peer Review File · Nature Communications]

Reviewer #1 (Remarks to the Author):

The manuscript presents an experiment on detection of single microwave photons with high efficiency (66%) and low dark-count probability (1%). The detector is realized with a microwave resonator coupled to superconducting flux qubit, with a special drive applied to the qubit, to realize a Raman transition triggered by the detected photon. For some values of the drive frequency and amplitude, the destructive interference suppresses the reflection of the input microwave, thus providing high detection efficiency.

The experiment is an important step in realizing high-efficiency detectors of microwave photons. The paper is well-written. In my opinion, the manuscript undoubtedly deserves publication in Nature Communications. Some optional comments are below.

1. The explanation of the idea on Page 6 is not very clear. For example, it is not obvious that equal radiative decay rates lead to impedance matching. It is also not quite clear why the "nesting" of the energy levels is important. I think it would be good to cite Refs. [24,25] earlier in the paragraph and/or, even better, explain the idea in a longer way in a special section in the Supplemental material.

2. It was not immediately clear to me that the quantum efficiency is normalized by the photons in the pulse outside the resonator (somehow I thought that n_{bar_s} relates to the photon number in the resonator). This should probably be emphasized.

3. For many applications requiring single microwave photons, the natural shaping of the signal is exponential (instead of Gaussian). I think it would be good to briefly comment if the detector would also work with exponential single-photon signals. (Does the rapid increase at the start of the exponential signal present a problem?)

4. It seems that the calibration of the input power is rather indirect. Is it possible to do the calibration in a more direct way (for example, using ac Stark shift)?

Reviewer #2 (Remarks to the Author):

This manuscript reports the first implementation of a single-photon detector for itinerant microwave photons, which is an important task and a long standing challenge for the field of circuit QED and microwave quantum optics.

This experiment is an excellent realization of a proposal initially laid out by the same author group more than two years ago (PRL 111, 153601 (2013)), where an impedance matched Λ -system, constructed from the dressed states of a superconducting flux qubit, deterministically absorbs a single incoming microwave photon in a 1D transmission line. The manuscript is clearly written from introduction and referencing to technical descriptions. I definitely recommend this work be published in Nature Communications.

There are competing ideas of single-photon detection in the microwave regime, such as using avalanching effects in Josephson circuits, or catching incoming photons into localized modes for subsequent detection. There have been note-worthy demonstrations of components for these approaches; progress from this author group in realizing their Λ -system has also been previously reported (PRL 113, 063604 (2014)), but the present work marks the first to show "clicks" of single-photon detection events. The performance of the detector is quite decent despite the relatively limited qubit coherence time, demonstrating the strong advantage of this Λ -system approach that really highlights coherent and deterministic light-matter interaction at the single quanta level. The authors also demonstrated and characterized a highly-practical reset protocol by reversing the coherent excitation process.

Before publication, I strongly suggest the authors address one main comment I have on the manuscript:

It is not clear to me at all from the main text how $P(0)$ is determined. Only after reading the supplementary material do I get a "hint" that it is probably just computed from the calibrated input power. I think the authors need to be more explicit in the main text in pointing this out for both clarity and (in some sense) honesty. I would also like to see some raw data of $P(e)$ and $P(0)$ in any of the plots, since for the weak coherent state, it seems we are dealing with a fairly small $P(e)$ and $P(0)$ in the measurements. Although readers can in principle calculate them from reported n_{bar} , it is always good to get a more tangible feel of these quantities from the authors first hand.

This comment also leads to two possible arguments against the manuscript:

First is that the reported efficiency is highly dependent on the input-power calibration, which in turn is dependent on fitting to a fairly complex theory. The reason that I trust the reported efficiency at all is because of Fig. 3b, which would be very different given a wrong calibration of n_{bar} . Therefore I

think it is important for the authors to emphasize this "self-consistency test" to their defense. And again, this is another good reason that they may want to plot $P(e)$ and $P(0)$ in Fig.3b as well.

The other argument is that this work only detects weak input coherent states rather than single input photons (which a quantum optics expert would likely demand) after all. It would be a more convincing single-photon detector if the authors can find a way to discriminate the instances when the coherent state is projected to $|0\rangle$ versus $|1\rangle$ on a single-shot basis, and post-select on $|1\rangle$. I don't see an obvious way how this can be done, and I don't insist this to be an absolutely essential element to make the claim of the paper, but the authors may want to acknowledge this limitation.

In addition, I have some minor comments:

1) On Page#10, Line#9, I don't quite understand why $P(e)$ is 0.49 without reset pulses. Could the authors explain?

2) There appears to be a typo on Page#10, the 3rd last line, the efficiency η should probably be 0.67 rather than 0.067.

3) In two places the T_1 of the qubit is claimed to be $\sim 0.7 \mu\text{s}$ and $\sim 900 \text{ ns}$. Is it due to different measurement conditions?

4) There is a newer work just appearing as a preprint on arXiv that also realizes single microwave-photon detection in a remote entanglement experiment. The authors may consider referencing it before final publication.

Reviewer #3 (Remarks to the Author):

The paper claims to demonstrate a device for detecting microwave photons with sufficient sensitivity to detect single microwave-photons (destructively).

The paper is well written, clear, concise and concerns original work. Generally the paper presents itself fairly in the context of the field (but see my comments below). Some of the authors (Tsai and Nakamura) are well-known leaders in the field. I could find no serious technical errors.

The development of single-photon detectors in the microwave regime is a goal shared by several different fields of research, including those involved in superconducting qubit development and astronomical detection. This gives the work a wider interest than just the superconducting quantum electronics community. The field is very active, see for example the significant recent progress in parametric amplifiers of the travelling wave type (included as Reference 8 of the paper).

The central claim of Inomato et. al., of having created a single-microwave-photon detection device is reasonably well justified and is reasonably convincing. The arguments relating to on-chip calibration of signal power and the constancy of the detection efficiency with varying signal power, together with other supporting information, help demonstrate that the device is sufficiently sensitive and indeed single-photon sensitive. This makes the device among the most sensitive to microwave photons of any yet devised.

The device discussed is not practical for widespread use, it is narrow-band, not yet sufficiently efficient and probably retains a too long dead time and too high dark count. It is also a complex instrument, the tuning process to achieve optimal operating parameters is a major task for experts. Travelling wave devices, for example, have some superiority in some desirable features over that proposed here, for example in their bandwidth and simplicity of operation, but need to be improved in other ways. However, the device discussed represents an early entry in the global competition to develop sufficiently sensitive and efficient detectors. As such, it serves as a proof of principle of a device and a technique and the work takes forward the state-of-the-art of microwave detection. While I think it unlikely that this work will strongly influence others to reconstruct such a device as a microwave photon-detector, knowledge of the successful implementation of the technique may inspire others to do different things with similar quantum processes (the use of qubits in cpw-cavities is quite widespread). For these reasons I would prefer to see the title changed to 'Detection of single microwave-photons...' as opposed to announcing a 'single microwave-photon detector'. Also, for these reasons, this report deserves dissemination in Nature Comms.

Perhaps the weakest aspect of the work is that it would have been better to test the device with an on-demand single-photon source rather than a coherent signal. However, such sources are themselves still a topic of research in the field and probably combining two such works asks too much at the present time, but it must come if the field of microwave-quantum-optics is to be properly established.

My other significant suggestion for the improvement of the paper is for the authors to consider including the discussion of the time constant of the impedance matched λ system in the main text rather than the supplementary material. I found that this section aided in my understanding of the operation of the device.

On a minor point, I found the symbol-sized error bars of Fig.3b confused the eye.

Response to Reviewer No.1

We wish to thank Reviewer for his/her careful reading and the recommendation for publication in Nature Communications.

The followings are answers to the Reviewer's comments.

The manuscript presents an experiment on detection of single microwave photons with high efficiency (66%) and low dark-count probability (1%). The detector is realized with a microwave resonator coupled to superconducting flux qubit, with a special drive applied to the qubit, to realize a Raman transition triggered by the detected photon. For some values of the drive frequency and amplitude, the destructive interference suppresses the reflection of the input microwave, thus providing high detection efficiency. The experiment is an important step in realizing high-efficiency detectors of microwave photons. The paper is well-written. In my opinion, the manuscript undoubtedly deserves publication in Nature Communications. Some optional comments are below.

Thank you very much for your high assessment on our work.

1. The explanation of the idea on Page 6 is not very clear. For example, it is not obvious that equal radiative decay rates lead to impedance matching. It is also not quite clear why the "nesting" of the energy levels is important. I think it would be good to cite Refs.[24,25] earlier in the paragraph and/or, even better, explain the idea in a longer way in a special section in the Supplemental material.

We cite Refs. 24 and 25 (now 25 and 26) earlier than in the previous version (line 15, page 6). Although we could describe details on how to implement the impedance-matched Λ system, that would just result in repeating the same explanations described in Refs. 24 and 25.

2. It was not immediately clear to me that the quantum efficiency is normalized by the photons in the pulse outside the resonator (somehow I thought that \bar{n}_s relates to the photon number in the resonator). This should probably be emphasized.

We emphasize that we use the mean photon number in the propagating signal pulses to define the detection efficiency of single-microwave photons (line 16, page 8). Also we give the definition of $P(0)$, that is $P(0) \equiv \exp(-\bar{n}_s)$ (line 14, page 8).

3. For many applications requiring single microwave photons, the natural shaping of the signal is exponential (instead of Gaussian). I think it would be good to briefly comment if the detector would also work with exponential single-photon signals. (Does the rapid increase at the start of the exponential signal present a problem?)

We briefly mention the case of the signal-photon pulse with exponential decay with a citation of our theory paper, Ref. 29, which describes the details of the shape dependence of the detection efficiency (line 16, page 9). As Reviewer pointed out, the rapid increase of the signal-photon pulse would affect the efficiency but the effect is not so significant, as we observe in Fig. 8 of Ref. 29.

4. It seems that the calibration of the input power is rather indirect. Is it possible to do the calibration in a more direct way (for example, using ac Stark shift)?

We measured an ac Stark shift as a function of a number of photons in the resonator. However, the shifted spectrum was not sharp and clear enough to carry out the calibration precisely. This is a reason why we did not use the conventional way to calibrate the input-microwave power. As we describe in Supplementary Note 3, we calibrate the input-microwave power in another way, which is based on experimentally determined parameters. We believe the method is reasonable and precisely calibrates the input-microwave power.

Response to Reviewer No.2

We wish to thank Reviewer for his/her careful reading and for the valuable comments and the recommendation for publication in Nature Communications.

The followings are answers to the Reviewer's comments.

This manuscript reports the first implementation of a single-photon detector for itinerant microwave photons, which is an important task and a long standing challenge for the field of circuit QED and microwave quantum optics.

This experiment is an excellent realization of a proposal initially laid out by the same author group more than two years ago (PRL 111, 153601 (2013)), where an impedance matched Λ -system, constructed from the dressed states of a superconducting flux qubit, deterministically absorbs a single incoming microwave photon in a 1D transmission line. The manuscript is clearly written from introduction and referencing to technical descriptions. I definitely recommend this work be published in Nature Communications.

There are competing ideas of single-photon detection in the microwave regime, such as using avalanching effects in Josephson circuits, or catching incoming photons into localized modes for subsequent detection. There have been note-worthy demonstrations of components for these approaches; progress from this author group in realizing their Λ -system has also been previous reported (PRL 113, 063604 (2014)), but the present work marks the first to show "clicks" of single-photon detection events. The performance of the detector is quite decent despite the relatively limited qubit coherence time, demonstrating the strong advantage of this Λ -system approach that really highlights coherent and deterministic light-matter interaction at the single quanta level. The authors also demonstrated and characterized a highly-practical reset protocol by reversing the coherent excitation process.

Thank you very much for your high assessment on our work.

Before publication, I strongly suggest the authors address one main comment I have on the manuscript:

It is not clear to me at all from the main text how $P(0)$ is determined. Only after reading the supplementary material do I get a "hint" that it is probably just computed from the calibrated input power. I think the authors need to be more explicit in the main text in pointing this out for both clarity and (in some sense) honesty. I would also like to see some raw data of $P(e)$ and $P(0)$ in any of the plots, since for the weak coherent state, it seems we are dealing with a fairly small $P(e)$ and $P(0)$ in the measurements. Although readers can in principle calculate them from reported n_{bar} , it is always good to get a more tangible feel of these quantities from the authors first hand.

Thank you for pointing out the ambiguity. $P(0)$ defined as $P(0) \equiv \exp(-\bar{n}_s)$, and as Reviewer correctly guessed, \bar{n}_s is determined from the input-power calibration. Following the Reviewer's suggestion, we added a new plot as Fig. 3b, in which the raw data of $P(|e\rangle)$ is presented. We also plot $P(0)$ calculated from \bar{n}_s in the same figure. The photon-detection efficiency η shown in Fig. 3c (former Fig. 3b) is calculated from $P(|e\rangle)$ and $P(0)$, now being easily confirmed by readers. We added the definition of $P(0)$ in the main text (line 14, page 8).

This comment also leads to two possible arguments against the manuscript: First is that the reported efficiency is highly dependent on the input-power calibration, which in turn is dependent on fitting to a fairly complex theory. The reason that I trust the reported efficiency at all is because of Fig. 3b, which would be very different given a wrong calibration of n_{bar} . Therefore I think it is important for the authors to emphasize this "self-consistency test" to their defence. And again, this is another good reason that they may want to plot $P(e)$ and $P(0)$ in Fig.3b as well.

We thank Reviewer for the important suggestion. We added a sentence (line 13, page 10) to emphasize the point according to the Reviewer's comment.

The other argument is that this work only detects weak input coherent states rather than single input photons (which a quantum optics expert would likely demand) after all. It would be a more convincing single-photon detector if the authors can find a way to discriminate the instances when the coherent state is projected to $|0\rangle$ versus $|1\rangle$ on a single-shot basis, and post-select on $|1\rangle$. I don't see an obvious way how this can be done, and I don't insist this to be an absolutely essential element to make the claim of the paper, but the authors may want to acknowledge this limitation.

Indeed, in every cycle in our experiment the detector says "0" or "1 (or more)". So, the weak coherent state input is projected to $|0\rangle$ versus $|1\rangle$ on a single-shot basis with some finite fidelity. We repeated the measurements 10,000

times for obtaining the statistics and determining the efficiency.

In addition, I have some minor comments:

(1) On Page 10, Line 9, I don't quite understand why $P(|e\rangle)$ is 0.49 without reset pulses. Could the authors explain?

This is because of the qubit relaxation. As we show in Fig. 4a, we first apply a π pulse to excite the qubit. We wait for ~ 410 ns before applying a readout pulse when we do not apply any reset pulse. Therefore, the qubit naturally decays and we obtain $P(|e\rangle) = 0.490 \pm 0.010$.

(2) There appears to be a typo on Page 10, the 3rd last line, the efficiency η should probably be 0.67 rather than 0.067.

Yes, thank you for pointing out the typo. We carefully checked our manuscript (especially, numbers) once again and we found another typo (line 13, Page 9), where, 5% should be 0.5%.

(3) In two places the T_1 of the qubit is claimed to be ~ 700 ns and ~ 900 ns. Is it due to different measurement conditions?

At the beginning of the cool-down for the experiment, the qubit showed $T_1 \sim 900$ ns, which was determined by averaging the values obtained in hundred identical measurements. However, after two months of the experiment at the base temperature of our fridge, T_1 suddenly dropped to ~ 700 ns. We always monitored T_1 before and after the photon-detection measurements, because the short T_1 affected the detection efficiencies. However, we did not understand why T_1 became shorter during the cool-down. One possible explanation is a coupling to a two-level fluctuator, which enhanced a qubit decay.

(4) There is a newer work just appearing as a preprint on arXiv that also realizes single microwave-photon detection in a remote entanglement experiment. The authors may consider referencing it before final publication.

Following the Reviewer's suggestion, we cite arXiv:1603:03742 in the introduction of the article (line 8, page 4).

Response to Reviewer No.3

We wish to thank Reviewer for his/her careful reading and the recommendation for publication in Nature Communications.

The followings are answers to the Reviewer's comments.

The paper claims to demonstrate a device for detecting microwave photons with sufficient sensitivity to detect single microwave-photons (destructively).

The paper is well written, clear, concise and concerns original work. Generally the paper presents itself fairly in the context of the field (but see my comments below). Some of the authors (Tsai and Nakamura) are well-known leaders in the field. I could find no serious technical errors.

The development of single-photon detectors in the microwave regime is a goal shared by several different fields of research, including those involved in superconducting qubit development and astronomical detection. This gives the work a wider interest than just the superconducting quantum electronics community. The field is very active, see for example the significant recent progress in parametric amplifiers of the travelling wave type (included as Reference 8 of the paper).

The central claim of Inomata et. al., of having created a single-microwave-photon detection device is reasonably well justified and is reasonably convincing. The arguments relating to on-chip calibration of signal power and the constancy of the detection efficiency with varying signal power, together with other supporting information, help demonstrate that the device is sufficiently sensitive and indeed single-photon sensitive. This makes the device among the most sensitive to microwave photons of any yet devised.

Thank you very much for your high assessment on our work.

The device discussed is not practical for widespread use, it is narrow-band, not yet sufficiently efficient and probably retains a too long dead time and too high dark count. It is also a complex instrument, the tuning process to achieve optimal operating parameters is a major task for experts. Travelling wave devices, for example, have some superiority in some desirable features over that proposed here, for example in their bandwidth and simplicity of operation, but need to be improved in other ways. However, the device discussed represents an early entry in the global competition to develop sufficiently sensitive and efficient detectors. As such, it serves as a proof of principle of a device and a technique and the work takes forward the state-of-the-art of microwave detection. While I think it unlikely that this work will strongly influence others to reconstruct such a device as a microwave photon-detector, knowledge of the successful implementation of the technique may inspire others to do different things with similar quantum processes (the use of qubits in cpw-cavities is quite widespread). For these reasons I would prefer to see the title changed to 'Detection of single microwave-photons...' as opposed to announcing a 'single microwave-photon detector'. Also, for these reasons, this report deserves dissemination in Nature Comms.

While Reviewer suggested changing it, we would like to keep the title of the manuscript as it is. We agree that the detection efficiency is not 100% and the operation requires some steps. However, the achieved quantum efficiency is already among the highest for microwave photons in the propagating mode, and is even higher than many optical single photon detectors commercially available. We believe it deserves the title. It would not prevent others from calling their own devices a single photon detector as well.

Perhaps the weakest aspect of the work is that it would have been better to test the device with an on-demand single-photon source rather than a coherent signal. However, such sources are themselves still a topic of research in the field and probably combining two such works asks too much at the present time, but it must come if the field of microwave-quantum-optics is to be properly established.

We agree with the Reviewer's comment. Now we are developing a single-photon source based on superconducting circuits. We hope to demonstrate the single-microwave-photon detection by combining the photon source and the detector presented in the manuscript.

My other significant suggestion for the improvement of the paper is for the authors to consider including the discussion of the time constant of the impedance matched Λ system in the main text rather than the supplementary material. I found that this section aided in my understanding of the operation of the device.

Following the Reviewer's suggestion, we added the discussion about the time constant of the impedance-matched Λ system (line 7, page 10).

On a minor point, I found the symbol-sized error bars of Fig.3b confused the eye.

We appreciate the Reviewer's comment. However, we have not found a better way to present the error bars. We hope the colors help to distinguish the data points.

A list of changes in the manuscript

The following is a list of changes based on the Reviewers' comments in our manuscript.

- “Temporal mode mismatch ...” and references [16] (line 6, page 4)
 - Suggestion by Reviewer No.2.
- Reference [25,26] (line 15, page 6)
 - As suggested by Reviewer No.1, we have cited the references [25,26] earlier.
- “ $P(0) \equiv \exp(-\bar{n}_s)$ ” (line 14, page 8)
 - Suggestion by Reviewer No.1 and No.2.
- “We emphasize that ...” (line 16, page 8)
 - Suggestion by Reviewer No.1.
- “approximately 0.5%” (line 13, page 9)
 - We have corrected the typo “approximately 5%”.
- “We theoretically confirm that ...” (line 16, page 9)
 - Suggestion by Reviewer No.1.
- “The characteristic response time ...” (line 7, page 10)
 - Suggestion by Reviewer No.3.
- Figure 3b (new) and its figure caption
 - As Suggested by Reviewer No.2, we have prepared Fig. 3b and changed a figure label of the former Fig. 3b to Fig. 3c.
- “Figure 3b shows ...” (line 11, page 10)
 - We have added an explanation of Fig. 3b.
- “ $\eta = 0.67 \pm 0.06$ ” (line 3, page 12)
 - As suggested by Reviewer No.2, we have corrected the typo “ $\eta = 0.067 \pm 0.06$ ”.

The following is a list of changes based on the manuscript checklist of Nature Communications.

- Divide the manuscript by headings of Results and Discussion (line 10, page 5 and line 7, page 12)
- Divide Results by subheadings (line 11, page 5, line 15, page 7, line 1, page 10, and line 2, page 11)
- Cite Supplementary items following the format of Nature Communications (line 13, page 5, line 1, page 6, line 3, page 9, line 10, page 9, and line 15, page 21)
- Describe definitions of error bars in Fig. 3 (line 5, page 22 and line 7, page 22)

Reviewer #2 (Remarks to the Author):

In the revised manuscript, the authors have taken sufficient consideration of the (optional) comments from all Reviewers. I do not have further questions, and I support its publication in Nature Communications.